# Comparative Study of the Restorative Effects of Forest and Urban Videos during COVID-19 Lockdown: Intrinsic and Benchmark Values

**DOI:** 10.3390/ijerph17218011

**Published:** 2020-10-30

**Authors:** Federica Zabini, Lorenzo Albanese, Francesco Riccardo Becheri, Gioele Gavazzi, Fiorenza Giganti, Fabio Giovanelli, Giorgio Gronchi, Andrea Guazzini, Marco Laurino, Qing Li, Tessa Marzi, Francesca Mastorci, Francesco Meneguzzo, Stefania Righi, Maria Pia Viggiano

**Affiliations:** 1Institute for Bioeconomy, National Research Council, 10 Via Madonna del Piano, I-50019 Sesto Fiorentino (FI), Italy; lorenzo.albanese@cnr.it; 2Pian dei Termini Forest Therapy Station, 2311 Via Pratorsi, I-51028 San Marcello Piteglio, Italy; ricerca@terapiaforestale.it; 3Section of Psychology—Department of Neuroscience, Psychology, Drug Research and Child’s Health (NEUROFARBA), University of Florence, 12 Via di San Salvi, I-50135 Firenze, Italy; gioele.gavazzi@unifi.it (G.G.); fiorenza.giganti@unifi.it (F.G); fabio.giovannelli@unifi.it (F.G.); giorgio.gronchi@unifi.it (G.G.); tessa.marzi@unifi.it (T.M.); stefania.righi@unifi.it (S.R.); mariapia.viggiano@unifi.it (M.P.V.); 4Department of Education, Languages, Intercultural Studies, Literatures, and Psychology (FORLILPSI), University of Florence, 12 Via di San Salvi, I-50135 Firenze, Italy; andrea.guazzini@unifi.it; 5Institute of Clinical Physiology, National Research Council, 1 Via Giuseppe Moruzzi, I-56124 Pisa, Italy; laurino@ifc.cnr.it (M.L.); mastorcif@ifc.cnr.it (F.M.); 6Department of Rehabilitation and Physical Medicine, Graduate School of Medicine—Nippon Medical School, 1-1-5 Sendagi, Bunkyo-ku, Tokyo 113-8603, Japan; qing-li@nms.ac.jp; 7Central Scientific Committee, Italian Alpine Club, 19 Via E. Petrella, I-20124 Milano, Italy

**Keywords:** anxiety, audio-visual stimulation, COVID-19, environmental enrichment, forest environments, forest therapy, lockdown, mental health, stress, quarantine

## Abstract

The prolonged lockdown imposed to contain the COrona VIrus Disease 19 COVID-19 pandemic prevented many people from direct contact with nature and greenspaces, raising alarms for a possible worsening of mental health. This study investigated the effectiveness of a simple and affordable remedy for improving psychological well-being, based on audio-visual stimuli brought by a short computer video showing forest environments, with an urban video as a control. Randomly selected participants were assigned the forest or urban video, to look at and listen to early in the morning, and questionnaires to fill out. In particular, the State-Trait Anxiety Inventory (STAI) Form Y collected in baseline condition and at the end of the study and the Part II of the Sheehan Patient Rated Anxiety Scale (SPRAS) collected every day immediately before and after watching the video. The virtual exposure to forest environments showed effective to reduce perceived anxiety levels in people forced by lockdown in limited spaces and environmental deprivation. Although significant, the effects were observed only in the short term, highlighting the limitation of the virtual experiences. The reported effects might also represent a benchmark to disentangle the determinants of health effects due to real forest experiences, for example, the inhalation of biogenic volatile organic compounds (BVOC).

## 1. Introduction

In March 2020, the outbreak of the COrona VIrus Disease 19 (COVID-19) pandemic turned into a major public health crisis and one of the greatest challenges we have to face. It constitutes a multidimensional stressor, affecting our whole society, single individuals, and families and involving physical personal risks, daily routines’ disruptions, uncertainty, social isolation, economic loss, unemployment risk, etc., [1,2]. Overall, emotional distress and increased risk for psychiatric illness associated with COVID-19 has emerged as further threats for society [3], and particular concerns were expressed about the effect of the epidemic on people with mental health disorders [4]. Some stressful factors derive from the “novel” nature of the illness and the consequent uncertainties related to the prognoses, treatment, modes of transmission, and effective prevention measures, as well to the government ability to cope with it (e.g., availability of resources for testing and treatment).

Moreover, the economic uncertainty worsened by the global pandemic crisis could affect psychological health and well-being through further stressors like unemployment risk, job uncertainty, debts, and recession [5].

Beyond the abovementioned stresses, the widespread social isolation, quarantine, and lockdown measures adopted in the attempt to contain the infection spreading caused high levels of psychological distress, unfavorably influencing mental health by compromising emotional and cognitive functioning. Many emotional effects were associated to the quarantine, including stress, depression, irritability, insomnia, fear, confusion, anger, frustration, and boredom [2]. Overall, the increase in mood disorders and higher anxiety levels seems to be a prevailing emotional response in quarantined people [6].

Many people were confronted with an unprecedented situation characterized by staying at home and changing completely their daily life. Along with other factors, also the lack of spending some quality time in natural environments might have also contributed to the global level of distress.

The responses and interventions to be undertaken in this crisis should be adequate to the levels of complexity and uncertainty of the pandemic. As far as psychological stress is considered, we suggest considering also the role of nature and its benefits on well-being perception.

Spending time in natural settings is a powerful body–mind experience that everybody knows and enjoys. Importantly, it is not only a subjectively enjoyable experience, but an astounding amount of supportive research demonstrated that being surrounded by nature generates significant benefits to physical and mental health [7,8,9,10].

In this context, a broad range of positive effects were related to forest exposure, ranging from an impact on mood, stress level, and other psychological variables, with focus on anxiety [11,12,13,14]. These beneficial effects were observed in both healthy subjects and clinical population. Even mental hospital patients with affective and psychotic disorders received useful psychological support from forest therapy practices [15]. Furthermore, the immersion in the quiet stillness of forest has potential health-promoting factors: Improvement of cardiovascular functions and hemodynamic neuroendocrine, metabolic, immune, inflammatory, oxidative, and electrophysiological indices [16].

A key research question focused on the identification of the key components of the forest setting that convey these health effects, as well as the attribution of the same effects to different components. A wealth of aspects contributes to make the contact with nature such a powerful tool for the human psychophysical well-being, while the effects of the forest on the human health are mediated by the use of visual, auditory, tactile, and olfactive senses [17,18].

Among the olfactive elements, long-lasting physiological health benefits were attributed to the inhalation of certain bioactive compounds residing in the forest atmosphere, including natural airborne microbiota and phytoncides, or biogenic volatile organic compounds (BVOCs), mainly monoterpenes (MTs), followed by sesquiterpenes [19]. For example, the MTs emitted by conifer trees were related to specific biological mechanisms, such as enhancing the numerosity and activity of natural killer cells (*α*-pinene and *d*-limonene) [20,21] and improving sleep duration and quality (*α*-pinene and 3-carene) [22].

It is also important to point out significant short-term psychological and physiological benefits deriving from the exposure to still images of forests, underlying the powerful effects triggered even by creating nature-based mental images [23]. Over the past decades, several studies confirmed restorative or stress-reducing effects by mediated exposure to natural environments through various types of substitute nature such as photos, videos, and virtual nature environments [24,25,26,27]. Just 6 min of nature exposure in a virtual reality setting can provide restorative effects and benefits on mood levels [28]. These findings are relevant also because not everyone has the possibility to experience natural environments at any time or at all and, therefore, has to face different “civilization” stressors derived from urbanization (sedentary lifestyle, over-exposure to digital media, feelings of isolation and disconnection, and anxiety) [29,30].

All these findings could contribute to promote the creation of virtual environments, from 2-D videos to immersive virtual reality, designed by landscape architects and urban planners, which could become important, nonmedical tools to improve the psychological and physiological health. These virtual tools might be particularly interesting and useful in specific contexts (e.g., hospitals) where nature-based immersive experiences are precluded or for people suffering from a motor disease.

Moreover, it might be important to compare the effects of virtual experiences with real forest immersion, also in order to disentangle the effects related to key features of natural forest settings. Indeed, real natural experiences unavoidably involve a multiplicity of stimulations, and recognizing the distinct effects produced by specific components of the forest environment, such as BVOCs, may be challenging.

The abovementioned evidence of health benefits led to the inclusion of forest therapy in national healthcare systems in a few countries in the Far East [31]. More recently, official recommendations for the promotion of forest therapy programs were issued in Western Europe, such as in the UK [32] and in Italy, where forest therapy was recognized as a socio-cultural service of the forest environments [33]. However, the exploitation of forest welfare services in Europe, aimed at tackling a number of social and demographic problems of European societies, is still a long way off, and the Forest Policy for Human Well-Being in South Korea was recently proposed as a useful model [34].

The focus of this study was to evaluate the possible beneficial effects of a virtual natural enrichment on daily life and specifically on anxiety levels. The circumstances related to the COVID-19 pandemic provided an extraordinary context for testing the effectiveness of the video exposure to the forest environments on stress and anxiety levels in healthy people forced by the lockdown to stay in limited and confined spaces in a condition of natural environment deprivation and lack of direct nature contact.

Testing during the COVID-19 quarantine was particularly important for two main reasons:The isolation caused by the lockdown might have raised the levels of anxiety (in individuals not affected by anxiety disorders).This extreme condition enabled us to study experimentally the anxiety level after excluding confounding factors of normal everyday life, such as social interaction or events or activities (sport, cultural, etc.).

This study aimed to explore whether an audio-visually presented forest-based video might boost a relaxing effect by reducing, in general, psychological activation and, specifically, state anxiety levels. A control condition was used, consisting of a video presenting an urban context.

The main hypotheses were that (1) the view of the short forest video affected levels of state and trait anxiety in healthy people forced to prolonged quarantine due to the lockdown measures and that (2) mediated nature experience can improve our capacity to recover in a stressful context, characterized by environmental isolation and lack of direct nature contact.

The pilot study intends to contribute to the evaluation of opportunities of mediated nature exposure as a way for mitigating anxiety disorders in a home-based environment during the quarantine period.

Moreover, evaluating the effect of the audio-visual exposure to forest environments on the psychological well-being could represent a benchmark, based on which the effects of other sensorial variables typical of real forest experiences, such as the inhalation of BVOCs, can be assessed and possibly disentangled.

## 2. Materials and Methods

### 2.1. Participants

Participants were recruited in Italy from the general population through announcements on social networks and mailing lists using a snowball strategy. A personal email containing a general explanation of the protocol was sent to individuals potentially interested in participating in the study who responded to the announcement.

All participants were ≥18 years of age and gave their informed consent to the procedure and the processing of personal data. All data were collected and processed anonymously. The study was performed according to the Declaration of Helsinki and was approved by the Ethical Committee of the University of Florence (No. 92811, July 2020).

A total of 90 participants were initially recruited and randomly assigned to the two experimental conditions described in Section 2.2. Due to dropouts (1 participant in the Forest condition and 14 participants in the Urban condition), the final sample consisted of 75 participants (59% female) with a mean age of 47.3 years (SD = 13.1). The population, by geographical area, was distributed as follows: central Italy (63%), northern Italy (28%), southern Italy (5%), or abroad (4%). None of them was living in Lombardy, the Italian region most affected by COVID-19.

### 2.2. Experimental Design

A parallel-group design was employed (Figure 1), where participants were randomly (based on participant entry) allocated in a 1:1 ratio to two different experimental conditions: (1) Watching an audio-video with forest environments (Forest condition), (2) watching an audio-video with urban environments (Urban condition).

The study was performed between 25 April and 3 May 2020. This period corresponded to the last week of Italy’s national strict lockdown due to COVID-19. On 4 May, the Italian government started to ease restrictions.

The video in the Forest condition consisted of forest environments located in the Apennine Mountains nearby the cities of Prato and Pistoia (Tuscany region). The video was shot in two different locations named Calvana (43°57′N, 11°10′E) and Acquerino (44°01′N, 11°00′E). Five different scenes without people were recorded, including coniferous and beech trees and water streams. Duration: 5.09 min, resolution 3686 by 2304 dots per inch (dpi), 25 frames per second (fps), audio 48,000 Hz.

The control video (Urban condition) consisted of urban environments. The video was shot in five different locations in downtown Prato, Tuscany, Italy. Five different scenes were recorded, comprising building scenes (office, front door, and window) without people. Duration: 5.10 min, resolution 3686 by 2304 dpi, 25 fps, audio 48,000 Hz. Both Forest video (S1) and Urban video (S2) are available in the Appendix A.

Before each video, a brief guide (8 s) invited the participant “to Breath, to Look at, to Listen to”, aimed at inducing a mindful attention mediated by the senses involved in that experience.

### 2.3. Self-Report Questionnaires

Measures included a demographic questionnaire and the State-Trait Anxiety Inventory Form Y (STAI) [35], acquired in baseline condition and at the end of the study, and the Part II of the Sheehan Patient Rated Anxiety Scale (SPRAS) [36], collected every day immediately before and after watching the video. The STAI is a widely used measure of state (STAI-Y1) and trait (STAI-Y2) anxiety consisting of 40 items, 20 each regarding state and trait, respectively. Each item is measured on a 4-point Likert scale, with a score of 1 referring to “Not at all” and a score of 4 indicating “Always”. For each subscale, scores may range from 20 to 80, with a high score indicating higher levels of anxiety. The STAI-Y1 contains items describing how the individual feels at the time of the assessment. The STAI-Y2 contains items related to general anxiety proneness, considered as a relatively stable characteristic of the personality, which reflects how the individual tends to perceive stimuli and environmental situations as dangerous or threatening.

The SPRAS Part II is an 11-item scale that elicits anxiety symptoms that occur in response to danger, stress, or a phobic stimulus [37]. The questionnaire enabled us to evaluate how well the subject could perceive his/her psychophysiological state during the assessment (e.g., “heart beating hard and quick”, “hand sweating”, etc.). Each item was scored by the subject from 0 to 4.

### 2.4. Procedures

The experimental procedure (Figure 1) was identical for both groups, except for the type of video assigned (Forest vs. Urban conditions). During the entire study week, each participant received a daily mail early in the morning with instructions and relative links to online questionnaires and videos.

The study started with a baseline assessment carried out in the weekend (25–26 April): Participants were requested to complete the demographic questionnaire and the STAI-Y1 and STAI-Y2. During five days (Monday to Friday), the two groups entered the intervention phase. Each participant was requested to watch the assigned video every day in the morning (between 8 a.m. and 12 p.m.) before starting her/his activities and before consulting news or newspapers. They were instructed to watch the video alone in a quiet, soundproof, and possibly isolated environment in the house. Moreover, they were invited to use headphones to better hear the sounds of the video.

Before and immediately after watching the assigned video, participants were required to fill the online SPRAS using the specific link. After the completion of the 5-day period, participants filled in again the STAI-Y1 and STAI-Y2 as post-intervention measures.

The 5-day trial period was chosen in order to guarantee the continuity of experiment and avoid introducing a discontinuity in habits, which would not have been possible if weekend days had been included in the period of the trial. Moreover, the uncertainty about the end of the strict lockdown period at the time of designing the experimental procedure made it difficult to plan a longer trial.

### 2.5. Data Analysis

The analysis was carried out only on cases that completed the SPRAS before and after viewing the assigned videos. Preliminary analyses compared the two groups across sociodemographic variables of interest: Age, gender, area of residence, presence of external space in the residence/could see a green area outside, education, presence of chronic diseases, regular practice of sports, previous experience with meditation or yoga as well as experience with COVID-19 (having relatives or close friends infected by the virus), impossibility to perform his/her sport inside the residence, forced isolation due to direct contact with COVID-19. Student’s *t* and χ2 tests were used, including pretreatment SPRAS values for each day.

A repeated-measure ANOVA with a condition as the independent variable and the pre- and post-treatment SPRAS measure as the dependent variable was performed for each day of the study. Based on previous literature, assuming a medium effect size (f = 0.30), with alpha = 0.05, a sample of 68 participants enabled a power of 0.83. The pre–post differences within each condition were assessed through paired-sample Student’s *t*-test (hereinafter indicated as “t”). The one-week, pre–post difference of the STAI-Y subscales (STAI-Y1 and STAI-Y2) were computed by means of paired-sample Student’s *t*-test.

## 3. Results

Descriptive analyses of sociodemographic data distinguished by the condition are reported in Table 1: No differences between the groups were observed for all the measured variables. No significant differences (*p* > 0.05) arose between the pretreatment SPRAS levels for any day.

Descriptive statistics of SPRAS data, including mean scores, standard deviation, and minimum and maximum values for each group, for each day of detection, and for each test administered are reported in Table 2.

Table 3 reports the results of mixed ANOVAs. The interaction effect was statistically significant for each day of the study, whereby the group assigned to the Forest condition had a lower value of SPRAS after watching the video (compared to pre-value), while the group assigned to the Urban condition had the same value (or higher) after watching the video.

Figure 2 shows the pre–post difference between SPRAS mean scores for each condition. In the Forest condition, we observed a reduction in the mean differences for each day of the study: Day 1 (t(40) = 3.91, *p* < 0.001), day 2 (t(40) = 3.58, *p* = 0.001), day 3 (t(40) = 3.42, *p* = 0.001), day 4 (t(40) = 2.46, *p* = 0.018), and day 5 (t(40) = 2.93, *p* = 0.006). In the Urban condition, there were no pre–post differences with the exception of day 3 (t(33) = −2.93, *p* = 0.006) and day 4 (t(33) = −2.93, *p* = 0.006).

Concerning STAI-Y, there were no significant one-week, pre–post differences in either condition for both state and trait anxiety measures (Table 4).

## 4. Discussion

In this study, we examined the effect of a 5-day intervention consisting of watching a video of natural environments on perceived anxiety levels in healthy individuals, during the last week of COVID-19 strict lockdown in Italy. The main finding was that a short exposition (5 min) to a forest video induced a self-perceived relaxing effect, as revealed by the reduction of SPRAS scores with respect to those reported immediately before watching the video. This result was specifically related to the forest video, while no effect was observed in the control group (urban video). The immediate perceived benefits induced after each daily viewing of the forest video was not associated with a prolonged effect. Indeed, no differences arose between conditions (Forest vs. Urban) after the 5-day intervention. Namely, in both groups (that were comparable in terms of initial scores) state and trait anxiety measures remained stable after one week.

Overall, our findings of virtual natural enrichment are in keeping with previous studies showing that viewing forest settings produced significantly more comfortable, calmer, and more refreshed feelings in the subjects than the city setting [38,39,40]. Based on these studies, walking in a forest setting and viewing forest landscapes can be effective in providing relaxation to people as compared to a city setting.

Interestingly, the short-term psychological benefits deriving from the audio-visual stimulation with forest imagery alone were consistent with the corresponding short-term psychophysical benefits observed after short walks in green suburbs, which were, in turn, comparable with benefits conveyed by similar walks in both coniferous and deciduous forests [41]. Apparently, phytoncides did not play a role in such walks because they were performed in fall when the deciduous trees were leafless and, thus, largely unable to emit BVOCs. Moreover, BVOCs emitted by plants in green suburbs easily undergo reactions with urban-generated nitrous oxides, leading to the generation of ozone and secondary organic aerosols [42]. In such situations, the visual experience is likely to be the most important determinant of the observed positive outcomes.

Several theoretical frameworks have been proposed to account for the beneficial effects of nature experience (considered as different degrees of exposure to natural environments: passive viewing of images, videos, or natural landscapes, and physical presence or immersion in the nature) on psychological health and well-being [43]. Two major, nonmutually exclusive theories are the stress reduction theory (SRT) [44] and the attention restoration theory (ART) [45].

According to SRT [44], natural environments with their sensory stimulations induce larger physiological and psychological relaxing effects compared to urban environments. In one of the experimental studies supporting this hypothesis, a group of healthy subjects were first exposed to a stressor (viewing a stressful movie about prevention of work accidents) for 10 min and then viewed one of six different natural and urban settings videos for 10 min. Physiological measures of the stress response (including electromyography, skin conductance response, heart rate, and blood pressure) were recorded along with self-rated affective states. All measures showed significantly higher speed of recovery from stress when subjects were viewing nature scenes with respect to urban scene conditions.

Another seminal study showed a restorative effect from the visual contact with nature (stay in a hospital room with a window view of a natural setting) on the speed of recovering from surgery [46]. Based on these studies, it was suggested that nature experience may promote our capacity to recover from a stressful event by helping to mitigate states of arousal within minutes from the exposure [47]. A restorative effect on stress level was recently observed in two studies aimed at determining the beneficial physiological effects of exposure to a virtual reality video showing forest resting [48] or high-biodiversity [49] environments. Similar to previous studies conducted in the framework of the SRT, the restorative effect was evaluated after experimentally inducing stress mental state (using a stressful mental arithmetic task).

In the present study, we exploited an exceptional situation, the Italian strict lockdown due to the COVID-19 outbreak, which represented, per se, a stressor. Indeed, several studies or surveys on COVID-19 conducted all around the word suggested that the lockdown and the quarantine might have generated or heightened emotional states in the form of increased perceived psychological distress and anxiety [50,51,52].

In most of the mentioned studies, the restorative effect of exposure to natural environments was evaluated by means of psychophysiological measures [47,48,49], whereas, in our study, only self-report questionnaires were employed; on the other hand, any assessment other than online questionnaires was unfeasible due to the lockdown restrictions. However, the SPRAS is composed of items that measure the severity of anxiety symptoms that are likely to be observed in the domain of a physiological individual’s response to stress [36,37].

The ART is another conceivable theoretical reference in which the results of this study can be framed [45]. This theory proposes that nature can renew attention and mental well-being after exerting mental energy, for instance, after spending sleepless nights studying for exams or working tirelessly on a project or assignment [53]. Specifically, the ART posits that the full immersion in natural environments would stimulate an effortless perception through all the sensory modalities: Sight, hearing, smell, etc., This would yield a sort of involuntary “attentional broadening”, producing a long-term restoration of our cognitive and affective status. Our study used audio-viewing condition and did not employ a full immersion in natural environments. Hence, we could speculate that the restoring effect of a virtual exposure to natural stimuli can benefit the short-term well-being (reduction of anxiety measured by SPRAS). However, in line with the ART, a full immersion in nature involving all sensory experience is needed to determine more lasting changes in anxiety (STAI).

This study was affected by some limitations, mainly due to the exceptional lockdown condition. COVID-19 outbreak represented a unique experience of social isolation and spatial confinement that is a no-laboratory, naturalistic model. In this context, it would not have been possible to evaluate all possible variables, which are generally well controlled in laboratory experiments, such as the environmental condition of video watching or other intervening factors potentially affecting the psychophysiological state of participants. Moreover, the peculiar context in which the research was carried out entailed a limitation in the possibility to recruit a larger sample size. Increasing the sample size in future research will allow applying more sophisticated statistical models that take into account directly the impact of potential moderating variables such as those listed in Table 1.

The extent of the benefits to mental health conveyed by the virtual nature experiences carried out in this study could represent a benchmark, upon which the distinct and likely enhanced benefits of real forest experiences can be assessed, as well as possibly to attribute such distinct benefits to specific predictors, including the exposure to BVOCs in the forest atmosphere. The accuracy of quantitative assessments of the additional value conveyed by real forest experiences can benefit from controlled, virtual experiences, especially when the effects generated by single or selected stimuli can be obtained. As well, real immersive experiences could be optimized in order to generate different or enhanced forest healing effects in comparison to virtual ones, such as by means of the inhalation of BVOCs.

Due to COVID-19 lockdown, this pilot study represents a naturalistic and unrepeatable condition. However, it is conceivable that the observed effect may be generalized in different social contexts characterized by similar characteristics of social isolation and spatial confinement, for example, hospitalized patients or inmates. Concerning the COVID-19 pandemic emergence, scientific literature is emerging about the management of possible further waves: The audio-visual exposure to forest-like videos, even though without long-lasting effects, might be useful to induce a temporary reduction of perceived anxiety, in the event that further lockdown periods are imposed [54,55].

## 5. Conclusions

This study examined the relaxation effect of audio-visual stimulation using a forest video, with an urban video as the control, in the absence of human–nature interactions, during COVID-19 quarantine. The results showed a short-term decrease of anxiety level after exposure to a forest video, while no long-term changes were observed. The virtual exposure to forest environments showed effective to reduce human stress levels, in people forced by lockdown to stay in limited spaces, in a condition of environmental deprivation and lack of direct nature contact.

Although virtual nature is unable to fully reproduce the effects of real nature [56,57,58,59], such as boosting immune functions due to exposure to phytoncides, immersive virtual nature (IVN) technologies could contribute to improve the physiological well-being of people who do not have direct access to nature [29,60], albeit only with short-term effects. Such approach could be useful for people whose contact with real nature is impeded or hazardous, such as individuals affected by physical disabilities or physiological and mental health disorders, including depression, anxiety, and psychosis, and inpatients [61].

Finally, the extent of benefits conveyed by affordable virtual nature experiences, such as the one reported in this study, could be used as a benchmark for the assessment and planning of real forest experiences, including, among the others, the exposure to BVOCs in the forest atmosphere. First, the results shown in this study could help in disentangling the effects generated by different elements concurring to real forest experiences. Second, any immersion in real forest environments, in order to really make sense, should convey higher or longer-lasting benefits to mental health and/or additional benefits to physiological health, in comparison with the ones reported in this study.

## Figures and Tables

**Figure 1 ijerph-17-08011-f001:**
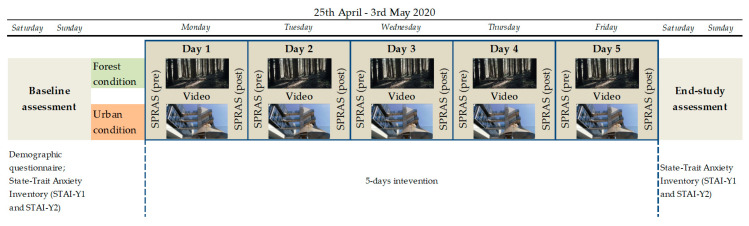
Experimental design.

**Figure 2 ijerph-17-08011-f002:**
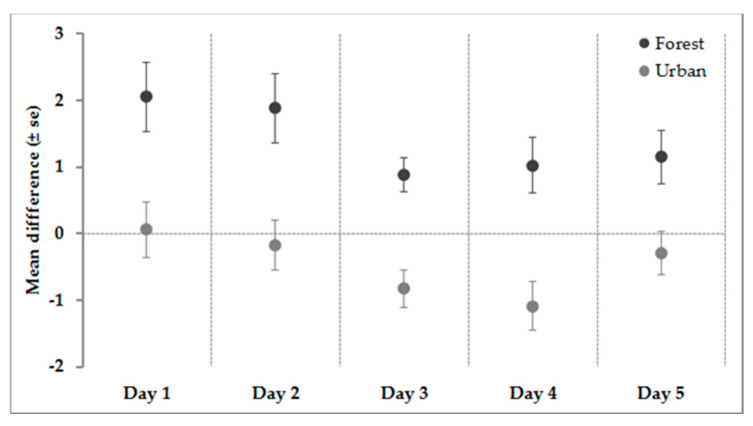
Mean differences between pre–post Sheehan Patient Rated Anxiety Scale (SPRAS) score for each day and condition. Values higher than 0 mean a reduction in SPRAS mean score after watching the video whereas values lower than 0 mean an increase in SPRAS mean score after watching the video.

**Table 1 ijerph-17-08011-t001:** Sociodemographic data.

Variable	Forest Condition (*n* = 41)	Urban Condition (*n* = 34)	Statistical Difference
Age (mean and SD)	45.6 (11.8)	49.3 (14.5)	t(73) = 1.22, *p* = 0.227
Gender	56% female	62% female	χ^2^ (1) = 0.24, *p* = 0.620
Residence	North: 25% Center: 63% South: 5% Abroad: 7%	North: 32% Center: 62% South: 6% Abroad: -	χ^2^ (3) = 2.95, *p* = 0.399
External space in the residence/could see a green area outside	95%	94%	χ^2^ (1) = 0.04, *p* = 0.847
Education	High school: 32% Italian Laurea (5 yrs) or higher: 68%	High school: 29% Italian Laurea (5 yrs) or higher: 71%	χ^2^ (1) = 0.05, *p* = 0.830
Experience with COVID-19	51%	38%	χ^2^ (1) = 1.26, *p* = 0.261
Forced isolation due to direct contact with COVID-19	5%	6%	χ^2^ (1) = 0.04, *p* = 0.847
Chronic disease	27%	21%	χ^2^ (1) = 0.40, *p* = 0.529
No regular practice of sports	32%	35%	χ^2^ (1) = 0.11, *p* = 0.743
Impossibility to perform his/her own sport inside the residence	65%	63%	χ^2^ (1) = 0.01, *p* = 0.908
Previous experience with meditation	44%	38%	χ^2^ (1) = 0.25, *p* = 0.620
Previous experience with Yoga	41%	29%	χ^2^ (1) = 1.17, *p* = 0.279

**Table 2 ijerph-17-08011-t002:** Descriptive statistics (mean scores, standard deviation, minimum and maximum values).

Day	Measure	Forest			Urban		
		*Mean*	*SD/SEM*	*Range*	*Mean*	*SD/SEM*	*Range*
**1**	Pre	5.20	5.22/.76	0–22	4.47	4.32/0.83	0–17
	Post	3.15	3.45/.59	0–15	4.41	4.14/0.65	0–14
**2**	Pre	5.05	5.11/0.67	0–19	3.18	2.96/0.73	0–11
	Post	3.17	3.25/0.50	0–13	3.35	3.09/0.55	0–15
**3**	Pre	3.61	3.63/0.54	0–18	3.12	3.23/0.59	0–17
	Post	2.73	2.81/0.55	0–13	3.94	4.20/0.60	0–22
**4**	Pre	3.85	4.17/0.58	0–18	3.00	3.03/0.63	0–14
	Post	2.83	2.73/0.51	0–11	4.09	3.77/0.56	0–17
**5**	Pre	4.54	6.33/0.85	0–21	2.82	4.13/0.93	0–21
	Post	3.07	3.76/0.59	0–17	3.12	3.79/0.65	0–16

**Table 3 ijerph-17-08011-t003:** Repeated measure ANOVA with condition as independent variable and SPRAS score as dependent variable for each day of the study.

Day	Main Effect of Condition	Main Effect of Pre-Post Treatment	Interaction Effect
**1**	F(1,73) = 0.08, *p* = 0.776	F(1,73) = 9.25, *p* = 0.003	F(1,73) = 8.25, *p* = 0.005
**2**	F(1,73) = 1.10, *p* = 0.299	F(1,73) = 6.45, *p* = 0.013	F(1,73) = 9.40, *p* = 0.003
**3**	F(1,73) = 0.21, *p* = 0.649	F(1,73) = 0.02, *p* = 0.887	F(1,73) = 19.91, *p* < 0.001
**4**	F(1,73) = 0.07, *p* = 0.789	F(1,73) = 0.01, *p* = 0.913	F(1,73) = 13.77, *p* < 0.001
**5**	F(1,73) = 0.64, *p* = 0.425	F(1,73) = 3.49, *p* = 0.066	F(1,73) = 7.89, *p* = 0.006

**Table 4 ijerph-17-08011-t004:** One-week, pre–post state (STAI-Y1) and trait (STAI-Y2) anxiety differences between Forest and Urban conditions.

Measure	Condition	One-Week Pre-Post Value Mean (sd, Range)	Paired-Sample Student’s t
STAI-Y1	Forest	42.5 (13.1, 21–62) – 43.1 (13.3, 22–72)	t(35) = −0.59, *p* = 0.556
	Urban	39.4 (10.6, 21–68) – 39.8 (11.4, 21–69)	t(27) = −0.32, *p* = 0.748
STAI-Y2	Forest	39.3 (10.4, 23–60) – 40.5 (11.2, 22–66)	t(35) = −1.19, *p* = 0.241
	Urban	39.0 (11.8, 21–61) – 39.1 (12.1, 20–67)	t(27) = −0.11, *p* = 0.915

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
