# Peer review of "Comparative Study of the Restorative Effects of Forest and Urban Videos during COVID-19 Lockdown: Intrinsic and Benchmark Values"

_ijerph, 2020, doi:10.3390/ijerph17218011_

Round 1
Reviewer 1 Report
A small methodological problem: in the groups of volunteers is only indicate the average of age, but not the several classes of ages. This is important because if there is a big group of elder people balanced by a big group of very young people the results are not coherent. So it is important to include a table of these ages.
The experimental design is very good, but the study was too short (only 5 days) and so the videos and preparation time for the participants. Too short time for taking all those conclusions, but it is possible to understand that the situation of COVID-19 was special moment and the study was placed at the end of worse period of it.
The paper is written in a good English and it is very intelligible.
Reviewer 2 Report
The study is interesting but does not show any innovative research. Rather it relies heavily on work undertaken previously by other researchers.
It is also doubtful that substantiated conclusions can be drawn from a study that was only 5 days long.
More value could be gained from a study conducted over a much longer period,
Reviewer 3 Report
A very useful and significant study. The paper itself is a well-conceptualized and written paper.
However, further improvements are suggested: 1. Lack of hypotheses support and better explaining the mechanisms of impact on health in a particular context of COVID-19 quarantine - specific isolation and its implications for mental health can be elaborated. Consequently, more current studies of COVID impact on mental health can be reviewed.
For instance:
Pfefferbaum, B., & North, C. S. (2020). Mental health and the Covid-19 pandemic. New England Journal of Medicine. Yao, H., Chen, J. H., & Xu, Y. F. (2020). Patients with mental health disorders in the COVID-19 epidemic. The Lancet Psychiatry, 7(4), e21. Rajkumar, R. P. (2020). COVID-19 and mental health: A review of the existing literature. Asian journal of psychiatry, 102066. Godinic, D., Obrenovic, B., & Khudaykulov, A. (2020). Effects of Economic Uncertainty on Mental Health in the COVID-19 Pandemic Context: Social Identity Disturbance, Job Uncertainty and Psychological Well-Being Model. International Journal of Management Science and Business Administration, 6(1), 61-74.Author Response
Please see the attachment

Reviewer 4 Report
I have carefully examined the study presented by the authors in which important and severe methodological and statistical analysis limitations emerge.
The study involved 75 participants placed in two different experimental conditions for a period of time between 25 April and 3 May. In Italy the lockdown ended on May 4th.
- First, it is noted how the resolution of the two videos is highly different. The resolution is reported as greater for the experimental condition and it should be specified how a high resolution allows a greater vision even of small details. The authors do not specify whether and how this data was observed in other previous studies and whether an effect related to video resolution was excluded in these studies.
- The SPRAS test was administered daily before and after viewing the 5.10 minute video. The authors claim to have invited attendees to view the video alone and in a quiet environment. How were these variables controlled? In any case, the fact remains that the participants were not in the same place and therefore in the same environmental condition. There is a limit to the ecological validity. In a design such as the one described by the authors, it is necessary to be able to keep all the intervening variables under control in order to affirm that the differences in the anxiety scores are related to the independent variable.
- The authors report nothing on the problem of controlling the expectations of success, bias in the response processes, the level of acquiescence, social desirability of the participants which normally affect the nature of the response process provided to self-report instruments.
- How can the authors exclude the incidence of other variables before, during or after the administration of the test that are different and independent from those considered?
- Have other measures been taken into consideration such as physiological ones such as heart rate etc.? Nothing is reported to this effect in the manuscript.
- Another very important methodological condition that does not seem to have been observed: where were the videos seen, or through what medium were they viewed: through smartphones, tablets, or televisions? Depending on the instrument and the characteristics of the support through which the video was viewed, there are very different graphics cards, audio and color resolution as well as image sizes that could have been engraved. Therefore the participants were not in the same conditions, regardless of the experimental condition associated with them. In the absence of these descriptions, it cannot be said that the participants found themselves in the same condition to be able to affirm that the possible effects recorded derive from the experimental condition assigned to them.
- The authors do not explain what they derived and established the 5-day trial period from.
In the absence of some methodological elements, even the statistical part is at least partially invalid and in any case it is insufficient.
- The authors have administered a personal data sheet. These variables were not used in any way in data processing. The chi-square detects any differences in the distribution of variables, such as Age, gender, residence external space In the residence, education, experience with covid, forsed isolation, chronic disease, sports, previous experience with meditation / yoga. The chi-square does not exclude that there may be an effect of a variable that has equal distribution in the groups. From a clinical point of view, numerous studies have found that anxiety levels are associated and determined by many factors, including those identified by the authors but which have not been included in the analysis models at all. This lack is relevant if we consider that the authors limit themselves to carrying out simple repeated measurements.
- Other variables that normally affect anxiety levels were not considered, such as the use of medications or drugs, living alone or with other cohabitants (friends, partners, children, etc.) and this all the more so during the period of lock down. Social status has not been investigated and nothing is reported about the participants' possibility of continuing to work during the lockdown. Furthermore, the personal data sheet does not seem to have investigated whether in addition to the lock down, the participants experienced other stress factors independent of the pandemic and therefore it does not seem possible to exclude the intervention of other causes with respect to which it is not possible to record even temporary beneficial effects from the vision of video.
- What levels of state and trait anxiety are we talking about? Normal, clinical or psychopathological levels?
- The authors report the objectives but do not make the hypothesis explicit.
- The authors do not seem to have taken into consideration the intragroup effect linked to living with external spaces In the residence / Could see a green area outside and this could be relevant precisely in relation to the objectives that the authors have set. A similar consideration is believed to be made for previous mediation or yoga experiences that may have facilitated the participants to obtain a relaxation effect.
- Table 2 shows significant pre and post treatment effects on the first and second day only. The discussion of the results, which appears completely insufficient, does not explain this result.
- Sta-y test-retest shows no difference before and after treatment In both experimental conditions. Also in this case the explanation is insufficient. What are the starting status and trait anxiety scores of the participants?
- The manuscript reports the differences in the means and t scores, but not the mean scores, standard deviation, minimum and maximum values for each group, for each day of detection and for each test administered. How can the validity of the results be assessed if basic elements are missing in the presentation of the results?
- The last days of detection coincided with the end of the lockdown. These are the days when there are no pre-post treatment effects. How do the authors exclude that it was this factor (end of the lockdown) that affected the results?
- In the urban condition, effects are recorded only in the last 3 days and not in the first. How do the authors explain this result? The trend of the experimental group is the opposite: the effect is high in the first 3 days, while smaller in the last two. The authors do not put forward any explanation. So even seeing urban video leads to the same effects or does the results depend on other factors other than video? The analyzes carried out by the authors do not allow an answer to this question.
- Do any effects emerge among those who usually live in the city and have seen forest vs urban video? and viceversa? How long or years have the participants been living in the city, small town, suburb, etc?
There is no real description of the study's limitations.
The study offered by the authors needs major and major revisions. At present the manuscript is not acceptable but may be revised again following various changes if they have remedied the limits described.
Round 2
Reviewer 1 Report
Now the paper has improved and it can publish as it is.
Author Response
Response to Reviewer 4 comments
Point 1: Now the paper has improved and it can publish as it is.
Response 1: The authors appreciate very much the positive comment of the esteemed Reviewer, and thank her/him for the careful review and insightful comments and suggestions.
Reviewer 4 Report
It should be pointed out that unfortunately the authors, while responding to the observations made, did not actually carry out a real process of revision and criticism on their work. The requested reviews highlighted the enormous limitations of the study but despite this the authors remained firm in their positions. the authors state that many factors have not been controlled and therefore this limits the validity and reliability of the results concerning a very small sample. Type I and type II errors were not checked. Many of the variables they cited were not considered because the authors say (but do not demonstrate and do not show) that no differences emerge. The t-test is not the only way to process data. The Anova model used by the authors is an excessively poor model of analysis. Analyzes must be improved.
On the basis of the critical reviews given to the first review, the authors highlighted many limitations to their study which, however, were not sufficiently highlighted in their manuscript. In fairness the authors should point out the important limitations without minimization. Expects to view the further analyzes requested by including other basic control variables in their analysis models, as is normally done in most analysis models. .
Author Response
Response to Reviewer 4 comments
Point 1: It should be pointed out that unfortunately the authors, while responding to the observations made, did not actually carry out a real process of revision and criticism on their work. The requested reviews highlighted the enormous limitations of the study but despite this the authors remained firm in their positions. The authors state that many factors have not been controlled and therefore this limits the validity and reliability of the results concerning a very small sample.
Type I and type II errors were not checked. Many of the variables they cited were not considered because the authors say (but do not demonstrate and do not show) that no differences emerge. The t-test is not the only way to process data. The Anova model used by the authors is an excessively poor model of analysis. Analyzes must be improved.
On the basis of the critical reviews given to the first review, the authors highlighted many limitations to their study which, however, were not sufficiently highlighted in their manuscript. In fairness the authors should point out the important limitations without minimization. Expects to view the further analyzes requested by including other basic control variables in their analysis models, as is normally done in most analysis models.
Response 1: The authors are very grateful to the esteemed Reviewer for her/his careful review and insightful comments and suggestions.
We added some amendments to further highlight the limitations of the study, which are disseminated throughout the manuscript.
About the comment concerning the statistical analysis: given the context in which the study was carried out (the last week of lockdown in Italy), we were able to the recruit a number of participants which allowed providing a sufficient power level in the case of a repeated ANOVA only with a single between variable. For this reason, we have chosen to maintain the current statistical analysis because taking into account other potential moderating variables will result in an excessive number of subsamples with very few cases (and, in several subsamples, no cases). In the Discussion section, we added this limitation due to the small sample size.